# Factors Associated with Reported COVID-like Symptoms and Seroprevalence Data Matched with COVID-like Symptoms in Slums and Non-Slums of Two Major Cities in Bangladesh

**DOI:** 10.3390/healthcare11101444

**Published:** 2023-05-16

**Authors:** Abdur Razzaque, Tarique Mohammad Nurul Huda, Razib Chowdhury, Md. Ahsanul Haq, Protim Sarker, Evana Akhtar, Md Arif Billah, Mohammad Zahirul Islam, Dewan Md. Emdadul Hoque, Shehlina Ahmed, Yasmin H. Ahmed, Fahmida Tofail, Rubhana Raqib

**Affiliations:** 1Health System and Population Studies Division, International Centre for Diarrhoeal Disease Research, Bangladesh (icddr,b), Dhaka 1212, Bangladesh; 2Department of Public Health, College of Public Health and Health Informatics, Qassim University, Al Bukairiyah 52741, Saudi Arabia; 3Infectious Disease Division, International Centre for Diarrhoeal Disease Research, Bangladesh (icddr,b), Dhaka 1212, Bangladesh; 4Embassy of Sweden in Bangladesh, Gulshan 2, Dhaka 1212, Bangladesh; 5Population Fund (UNFPA), Dhaka 1207, Bangladesh; 6Foreign, Commonwealth & Development Office (FCDO), Dhaka 1212, Bangladesh; 7Bangladesh Health Watch, James P Grant School of Public Health, BRAC University, Dhaka 1213, Bangladesh; 8Nutrition and Clinical Services Division, International Centre for Diarrhoeal Disease Research, Bangladesh (icddr,b), Dhaka 1212, Bangladesh

**Keywords:** reported COVID-19, seroprevalence, slum, non-slum, Bangladesh

## Abstract

Objectives: To examine the levels and socio-demographic differentials of: (a) reported COVID-like symptoms; and (b) seroprevalence data matched with COVID-like symptoms. Methods: Survey data of reported COVID-like symptoms and seroprevalence were assessed by Roche Elecsys^®^ Anti-SARS-CoV-2 immunoassay. Survey data of 10,050 individuals for COVID-like symptoms and seroprevalence data of 3205 individuals matched with COVID-like symptoms were analyzed using bivariate and multivariate logistic analysis. Results: The odds of COVID-like symptoms were significantly higher for Chattogram city, for non-slum, people having longer years of schooling, working class, income-affected households, while for households with higher income had lower odd. The odds of matched seroprevalence and COVID-like symptoms were higher for non-slum, people having longer years of schooling, and for working class. Out of the seropositive cases, 37.77% were symptomatic—seropositive, and 62.23% were asymptomatic, while out of seronegative cases, 68.96% had no COVID-like symptoms. Conclusions: Collecting community-based seroprevalence data is important to assess the extent of exposure and to initiate mitigation and awareness programs to reduce COVID-19 burden.

## 1. Introduction

The severe acute respiratory syndrome coronavirus-2 (SARS-CoV-2) was first detected in Wuhan, China in late 2019, and within a short period of time, it spread all over the world. The World Health Organization (WHO) officially termed the disease as Coronavirus Disease 2019 (COVID-19) [1,2,3] and declared the outbreak of COVID-19 a “pandemic” on 11 March 2020. People infected with SARS-CoV-2 may experience either no symptom or mild to severe symptoms. Typical symptoms of common cold are usually observed in mild cases, but severe cases often required hospitalization, long hospital stay, and may result in multiorgan failure and increased mortality [4].

In countries with larger populations, although having more numbers of both cases and deaths, older people and male sex with various health complications suffered the most [5]. COVID-19 is general susceptibility [6,7] and is accompanied by a cluster of flu-like symptoms and life-threatening severe illnesses including acute respiratory distress syndrome, acute kidney injury, myocarditis, and organ failure [8,9].

Globally, as of 10 November 2021, there have been 250,715,502 confirmed cases of COVID-19, including 5,062,106 deaths; a total of 7,084,922,999 vaccine doses have been administered [10]. Many countries have experienced second and third waves, and the new variants have also been identified in several countries. Fortunately, the COVID-19 situation is improving in many countries with the availability of vaccines. In Bangladesh, COVID-19 case was first reported on 8 March 2020 [11] and until 5 November 2021, there have been 1,570,485 confirmed cases of COVID-19 with 27,887 deaths; 72,712,400 vaccine doses have been administered [12].

It is evident that people living in urban slums are at risk as the environment in the slums is favorable for disease transmission with the overcrowded living conditions and limited access to the public health infrastructure [13]. Globally, about one billion people live in slum settlements, and more than half of them are in Asia [14]. SARS-CoV-2 spreads from droplets emitted through breathing, talking, coughing, and sneezing [15]; social distancing is more an aspiration than an attainable reality, particularly in the slum environment [16].

It is evident that people may be confused, anxious, or fearful about COVID-19. The fear has led to social stigma among certain groups, particularly for poor people. Moreover, there was a strong belief among the slum dwellers that COVID-19 is a disease of rich people [17]. Thus, there could be substantial underreporting of COVID-like symptoms. In fact, SARS-CoV-2-infected people can be symptomatic as well as asymptomatic and can transmit the virus irrespective of their symptoms [18]. As some symptoms of common flu and COVID-19 are similar; testing is needed to determine the etiology of illness and to confirm the diagnosis [19]. The objectives of the study were to examine the levels and socio-demographic differentials of: (a) reported COVID-like symptoms; and (b) seroprevalence data matched with COVID-like symptoms.

## 2. Materials and Methods

### 2.1. Setting

The data for this study came from selected slums and non-slum areas of Dhaka, the capital, and Chattogram, a seaport city. For this study, relatively large slums were purposively selected; four slums in and around Dhaka city (Korail, Mirpur, Dhalpur, and Ershad Nagar), and two slums in Chattogram city (Shaheed Lane and Akbar Shah Kata Pahar), using Slum Census Report-2014 [20]. Areas surrounding these slums were selected as non-slum areas. A slum is defined here as a cluster of compact settlements of five or more households that generally grow haphazardly in an unhealthy environment using poor construction material (for detail, BBS 2015 [20]). On the other hand, the non-slum areas were selected to include the middle-class, and the households were selected based on the structure of the house (wall and roof constructed with bricks; could be a multi-storied building, but no more than one security guard in the building).

### 2.2. Study Design

For selecting the sample, cluster random sampling procedure was followed. After mapping the study area, households were divided into clusters (equal size) and clusters were selected randomly to get the required sample size (Dhaka: 15 clusters from slum, 12 clusters from non-slum; Chattogram: 12 clusters from slum, 8 clusters from non-slum). Urban Health and Demographic Surveillance System (UHDSS) sampling frame was used for selecting the household in Dhaka slums [21]. For non-slum of Dhaka, and slum and non-slum of Chattogram, the household listing was done for the selected clusters before actual data collection began.

### 2.3. Survey Data and Sample Collection

For Dhaka slums, the fieldwork was undertaken by icddr,b; while for non-slum areas of Dhaka, and slum and non-slum areas of Chattogram, a private survey research firm was contracted for data and sample collection. Two weeks training program was organized by International Centre for Diarrhoeal Disease Research, Bangladesh (icddr,b) for data collection in Dhaka slums. The field coordinator of UHDSS coordinated the training program under the guidance of the investigators, while respective trainers and field workers participated. The training program had three components: questionnaire survey, nutritional measurement, and blood specimen collection. For collecting data from Dhaka non-slum and Chattogram slum and non-slum, the survey research firm organized the training program; however, the field coordinator of UHDSS was involved as a trainer to train the trainers. Subsequently, the trainers of the survey research firm trained their field workers and the duration of the training program was also two weeks. The icddr,b quality assessment team was responsible to check the data quality during the data collection period. The data were collected using Tablet/Android-based electronic questionnaire; the data collection continued for 5 months (October 2020 to February 2021) when vaccination against SARS-CoV-2 had not been administered in Bangladesh.

The study used survey data (questionnaire-based), as well as seroprevalence data of a sub-sample (Figure 1). The survey data were collected by interviewing the household head or any adult household member. Participants were enrolled in the study after obtaining informed consent/assent. We excluded those who were not aware of the national COVID-19 advice from the analyses and willingly did not report any of the COVID-like symptoms. Individual-level data (3965 households with 10,050 population, age 10 or more years) includes age, sex, education, occupation, and COVID-like symptoms. For reported COVID-like symptoms, the enumerators first asked the respondents (18 years or more and able to provide reliable information) whether any family member including him/her have had fever, cough, sore throat-like symptoms in the last 6 months. The respondents were then asked whether they had the following symptoms for each of the family member: fever or chills, cough, sore throat, congestion or runny nose, shortness of breath or difficulty in breathing, fatigue, muscle or body aches, headache, loss of taste or smell, nausea or vomiting and diarrhea [8]. This procedure of data collection along with adequate probing were followed to minimize reporting bias.

After household selection, all the eligible members aged ≥10 years, were also invited to participate to donate blood. Whenever a household head was unwilling to participate in the study, or the house was found locked or otherwise vacant, household was excluded and the next selected household in the cluster was approached. For seroprevalence data, blood sample was collected from 3220 individuals and analyzed 3205 (2436 adults aged ≥18 years and 769 children of 10–17 years) under the inclusion criteria. Single venous blood samples (7.5 mL) were collected in trace element-free heparinized tubes in the household and were delivered to the Laboratory within 2–3 h. Plasma was separated and stored at −80 °C until analysis. Confirmed COVID-19 cases included those who were identified as positive by PCR for SARS-CoV-2 in nasopharyngeal samples (by checking test report).

The Elecsys^®^ Anti-SARS-CoV-2 assay was used to determine the nucleocapsid (N) antigen specific antibodies (IgM and IgG) against SARS-CoV-2 in plasma on Cobas-e601 immunoassay analyzer (Roche Diagnostics GmbH, Mannheim, Germany) indicating recent or prior infection. Both methods provide a result of total antibody response (combination of both IgG and IgM antibodies) to N protein [21]. Based on the antibody cut-off index (COI), the serological response to SARS-CoV-2 is categorized as reactive (COI ≥ 1.0, seropositive) and non-reactive (COI < 1.0, seronegative) (for detail, Raqib et al., 2022 [22]). However, this study considered a combination of at least three symptoms (having fever, sore throat, and cough) for reporting COVID-like symptoms. This approach may reduce the bias to some extent. Moreover, during the pandemic, a great majority of participants exhibiting at least three or more flu-like symptoms were detected positive for SARS-CoV-2 infection [23].

The seroprevalence data matched with COVID-like symptoms were classified as reactive and non-reactive. The reactive cases with COVID-like symptoms were termed as symptomatic and the reactive cases without COVID-like symptoms were termed as asymptomatic. The non-reactive cases with COVID-like symptoms were termed as symptomatic-seronegative and the non-reactive cases without COVID-like symptoms were termed as asymptomatic-seronegative.

### 2.4. Outcome Variable

To examine the socio-demographic differentials of reported COVID-like symptoms, the outcome variable was either presence or absence of COVID-like symptoms (yes or no), while to examine seroprevalence data matched with COVID-like symptoms, the outcome variable was whether seropositive data matched with presence of COVID-like symptoms (yes or no).

### 2.5. Independent Variables

The independent variables were: age group (10–17, 18–59, and 60 years or more), sex (male and female), area (slum and non-slum), city (Dhaka and Chattogram), education (years of schooling) (0 years, 1–4 years, 5–9 years, and 10 or more years), occupation (not-working/unemployed, service, business, self-employed, housewife, student, and others), household income (BDT < 15,000, 15,000–19,000, and 20,000 or more), whether income reduced during COVID-19 (yes or no).

### 2.6. Statistical Analyses

For the descriptive analyses, proportions/rates were calculated. We observed the association between the reported COVID-like symptoms and seroprevalence using the Chi-square analyses. For both bivariate and multivariate logistic analyses: (a) individuals who suffered (10,050 individuals) from fever, cough, or sore throat were coded as “1”, and those who did not suffer from any of these symptoms were coded as “0”; and (b) 3205 individuals who were seroprevalence and had COVID-like symptoms were coded as “1”, otherwise coded as “0”; odd ratios along with 95% confidence interval were obtained and interpreted. Bivariate logistic analyses were conducted for both COVID-like symptoms and seroprevalence data matched with COVID-like symptoms to observe the crude effects of the independent variables.

## 3. Results

For COVID-like symptoms data, there was more slum than non-slum population (54.4% vs. 45.6%), and more population from Dhaka than Chattogram (53.9% vs. 46.1%) (Table 1). About 19% population were from age range 10–17 years, over 73% in age range 18–59 years, and the rest in age range 60 years or more. There were more females than males; however, the male-female difference was higher for non-slum than slum. As expected, there were differences in education, occupation, and household income between slum and non-slum; however, almost a similar percent of households from slum (71.5%) and non-slum (73.2%) areas suffered from income reduction during the pandemic. For seroprevalence data matched with COVID-like symptoms, the pattern by socio-demographic characteristics is almost similar to those observed for the COVID-like symptoms except those of income reduction.

For combined area, the reported COVID-like symptoms were higher in non-slum than slum area (35.1% vs. 29.4%) higher in Chattogram than Dhaka (34.3% vs. 30.0%) (Table 2). The COVID-like symptoms were higher in adults (33.1%) and elderly (33.3%) than adolescents (27.3%). Male and female had almost similar COVID-like symptoms (32.8% vs 31.5%). Those with longer years of schooling (for example, ≥10 years) had higher COVID-like symptoms than those with no-schooling (37.4% vs. 26.1%). Those who belong to unemployed/not-working category had lower COVID-like symptoms than all other occupation (for example, service holder) categories (25.8% vs. 36.7%). Higher-income group (for example, 20,000+ BDT) had slightly higher COVID-like symptoms than lower-income group (33.0% vs. 30.8%); reduction in household income had a slightly higher COVID-like symptoms than those with no reduction (32.7% vs. 30.2%).

For combined area, the seroprevalence data matched with COVID-like symptoms were higher in non-slum than slum participants (32.8% vs. 22.9%) (Table 2). Adults (28.3%) and elderly (25.3%) had higher matched cases than adolescents (19.2%). Males had a slightly higher matched cases than female (26.3% vs. 25.6%). Longer years of schooling (for example, ≥10 years) had a higher match cases than illiterate (32.5% vs. 20.3%). The unemployed/not-working category had a lower match than all other occupation (for example, service holder) categories (18.0% vs. 32.5%). Higher-income group (for example, 20,000 + BDT) had a slightly higher match than lower-income (27.6% vs. 24.2%). Reduction of household income had a slightly higher match cases than no reduction (26.1% vs. 25.9%).

Overall, more than 92% population having reported COVID-like symptoms had some kinds of treatment, 2% were hospitalized, and nearly 7% were tested for COVID-19; these were slightly higher in non-slum than slum (Figure 2). Of those who tested for COVID-19, none reported positive in slum, but 3.4% reported positive in non-slum.

Of the 2200 seropositive individuals, 37.77% were symptomatic and 62.23% were asymptomatic (Table 3). The symptomatic-seropositive prevalence was higher in the non-slum areas (53.26%) than slum areas (31.98%) and asymptomatic-seropositive was lower in the non-slum areas (46.74%) than slum areas (68.02%). The prevalence of asymptomatic-seronegative was higher in the slum areas (70.68%) than the non-slum areas (66.04%). Out of 1005 seronegative individuals, 68.96% showed no COVID-like symptoms.

Without adjustment, we found that the odds of COVID-like symptoms was higher in Chattogram city than Dhaka (OR = 1.21; 95% CI: 1.12, 1.32; *p* < 0.001), higher for non-slum than slum dwellers (OR = 1.30; 95% CI: 1.19, 1.41; *p* < 0.001), people of higher age groups (for example age 18–59) than people with age 10–17 (OR = 1.31, 95% CI: 1.17, 1.47; *p* < 0.001), having longer years of schooling (for example, ≥10 years) than no-schooling (OR = 1.70; 95% CI: 1.50, 1.92; *p* < 0.001), for working class (for example, service holder) than unemployed/not-working (OR = 1.67; 95% CI: 1.40, 1.99; *p* < 0.001), for households affected by income reduction than those without reduction (OR = 1.12; 95% CI: 1.02, 1.23; *p* = 0.016). (See Appendix A). For the slum population, the pattern was almost same except for income. While the pattern was same for slum non-slum except for city, occupation, income, and income reduced.

After adjusting other covariates, the odds of COVID-like symptoms was higher in Chattogram city than Dhaka (aOR = 1.13; 95% CI: 1.03, 1.23; *p* = 0.007), higher for non-slum than slum dwellers (aOR = 1.19; 95% CI: 1.05, 1.36; *p* = 0.009), people having longer years of schooling (for example, ≥10 years) than no-schooling (aOR = 1.59; 95% CI: 1.36, 1.86; *p* < 0.001), for working class (for example, service holder) than unemployed/not-working (aOR = 1.52; 95% CI: 1.26, 1.84; *p* < 0.001), for households affected by income reduction than those without reduction (aOR = 1.14; 95% CI: 1.03, 1.26; *p* = 0.008), while for households with higher income (for example, BDT 20,000 or more) had lower risk than those with lower income (BDT < 15,000) (aOR = 0.84; 95% CI: 0.74, 0.95; *p* = 0.006) (Table 4). The pattern was almost the same for slum except income reduction whereas for non-slum higher schooling and service holder had significant risk factors.

Without adjustment, we found that the odds of COVID-like symptoms was higher for non-slum than slum dwellers (OR = 1.64; 95% CI: 1.39, 1.93; *p* < 0.001), people of higher age groups (for example age 18–59) than people with age 10–17 (OR = 1.66, 95% CI: 1.35, 2.03; *p* < 0.001), having longer years of schooling (for example, ≥10 years) than no-schooling (OR = 1.87; 95% CI: 1.49, 2.36; *p* < 0.001), for working class (for example, service holder) than unemployed/not-working (OR = 2.20; 95% CI: 1.61, 2.99; *p* < 0.001) (See Appendix A). For the slum population, the pattern was almost same except for city, sex, and income. While for the non-slum population, the odds of seroprevalence data matched with COVID-like symptoms was lower for Chattogram than Dhaka (OR = 0.70; 95% CI: 0.51, 0.95; *p* = 0.024) and higher for those aged 18–59 years than those aged 10–17 years (OR = 1.51; 95% CI: 1.06, 2.16; *p* = 0.023).

After adjusting other covariates, the odd of having both seropositivity and COVID-like symptoms was higher for non-slum than slum dwellers (aOR = 1.67; 95% CI: 1.31, 2.14; *p* < 0.001), and for working class (for example, service holder) than unemployed/not-working (aOR = 1.80; 95% CI: 1.28, 2.52; *p* = 0.001) (Table 5). The pattern was almost same for slum but none of the risk factors was significant except those who were in service, business, or self-employed than unemployed/not working group. Contrarily, none of the risk factors was significant for non-slum area.

## 4. Discussion

The study demonstrated that the odds of COVID-like symptoms were significantly higher for Chattogram city than Dhaka, non-slum than slum dwellers, those longer years of schooling than no-schooling, working class than unemployed/not-working, households affected by income reduction than those without reduction, and lower for households with higher income than those with lower income group. For the matched cases, the odds of seropositivity and COVD-like symptoms were significantly higher in non-slum than slum dwellers, those longer years of schooling than no-schooling, and working class than unemployed/not-working. In the sub-population who were included in the serosurvey, 68.0% were seropositive. Of them, about one-third were symptomatic, while rest (two-third) were asymptomatic.

In this study, seropositive cases were largely asymptomatic (62.4%), and the rest suffered from mild to moderate disease (37.6%), which resulted in lower reporting of COVID-like symptoms. Growing data suggest that asymptomatic infections that largely remain undetected and can transmit the SARS-CoV-2 virus, that may act as the driver of the pandemic [24]. However, it is difficult to distinguishing people who are asymptomatic throughout the infection course, from those who are pre-symptomatic and would develop symptoms later; there is a disagreement about the extent (20–80%) to which asymptomatic infection can contribute to disease transmission [25,26]. The present study found higher asymptomatic cases in slums (68.23%) than non-slum areas (46.74%). Higher COVID-like symptoms among non-slum participants could be due to better reporting, as there were more educated people with greater awareness about COVID-19 symptoms living in non-slum than slum areas. Akhter et al. (2022) reported that slum dwellers believe that COVID-19 is a disease of rich people and not of the poor; a COVID-like symptoms may have been wrongly considered as common flu by the slum dwellers leading to under reporting [17]. Moreover, our study did not find any RT-PCR-confirmed COVID-19 positive cases among slum dwellers, while 3.4% positive cases were found in non-slum areas during the reference period. Respiratory symptoms were reported by the unemployed and retired persons, and cold-related symptoms were reported by the educated people and undergraduates [27]. Mouliou et al. (2022) reported that COVID-19 diagnosis has appeared to be ambiguous, and physicians need to correlate medical history, medical examination, potential extrapulmonary manifestations, along with laboratory and radiologic data, for an accurate COVID-19 diagnosis [28].

The present study reported similar level of COVID-like symptoms among male and female, and a similar pattern was also found for matched seropositivity with COVID-like symptoms. A similar finding was also reported by Nazneen et al. (2020) using community-based data [29]. However, the molecular test-based data of the Government of Bangladesh showed a high prevalence of COVID-19 (RT-PCR) among males compared to females [30], which might be linked to the prioritization of men in healthcare-seeking practices in the Bangladeshi context [31,32]. Thus, men might have sought more care from the government-based COVID-19 reporting sources. It is also possible that females had milder symptoms and thus did not go for COVID testing.

The main advantage of this study is that it was a community-based study that provided better reflection of the burden of COVID-19 rather than the findings from the hospital/clinic-based studies. Moreover, seroprevalence data matched with COVID-like symptoms allowed to identified symptomatic and asymptomatic cases. The limitation of the study is that data on COVID-like symptoms were collected with a relatively longer reference period (October 2020–February 2021); since the beginning of COVID-19 (March 2020) and likely to suffer from recall bias; however, interviewers were trained to collect symptoms by minimizing recall bias. The serosurvey was carried over a long period (~5 months) during which the transmission levels could change making it difficult to estimate the exact prevalence [22]). However, during the period of December 2020 to February 2021, the rate of active infection (RT-PCR) was low and thus may have minimally affected the survey. COVID-like symptoms are not accurate to diagnose COVID-19 disease, and we are aware that a great majority of people were asymptomatic [23]. However, during the pandemic the overwhelming burden of SARS-CoV-2 tended to overshadow the presence of other respiratory pathogens. Any mild flu-like symptom was found to be positive for SARS-CoV-2 [28]. Moreover, the perception of flu-like symptoms was also treated as COVID.

## 5. Conclusions

Symptoms of any disease is usually not disease-specific. Many symptoms overlap between different diseases and makes it difficult to differentiate between diseases. We have shown that broadly, data of symptoms of COVID-19 disease (of pandemic nature) matched and were in the same direction as that of the serosurvey, being higher in slums, lower in children, and similar in males and females. The novel disease COVID-19 had similarities with common flu and other acute respiratory infections. Our findings suggested that collecting symptoms of COVID-19 during the pandemic had some usefulness when extensive laboratory-based testing is not possible in crisis situation. As both symptomatic and asymptotic SARS-CoV-2 seropositive participants can transmit the virus and infect people, collection of community-based symptom data to determine trend and spread of infection may be useful in predicting and preventing future transmission of the disease by introducing mitigation and awareness programs to reduce COVID-19 infection.

## Figures and Tables

**Figure 1 healthcare-11-01444-f001:**
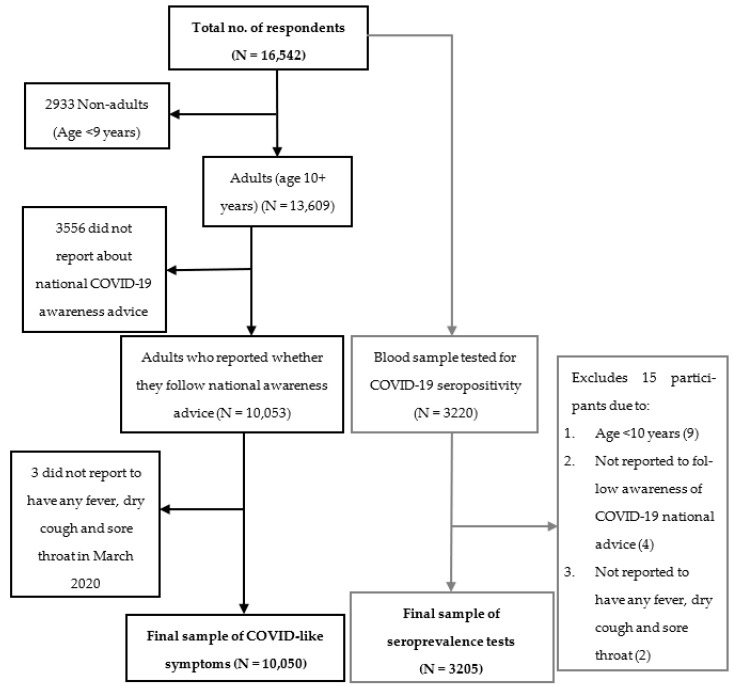
Sample selection for both COVID-like symptoms and seroprevalence.

**Figure 2 healthcare-11-01444-f002:**
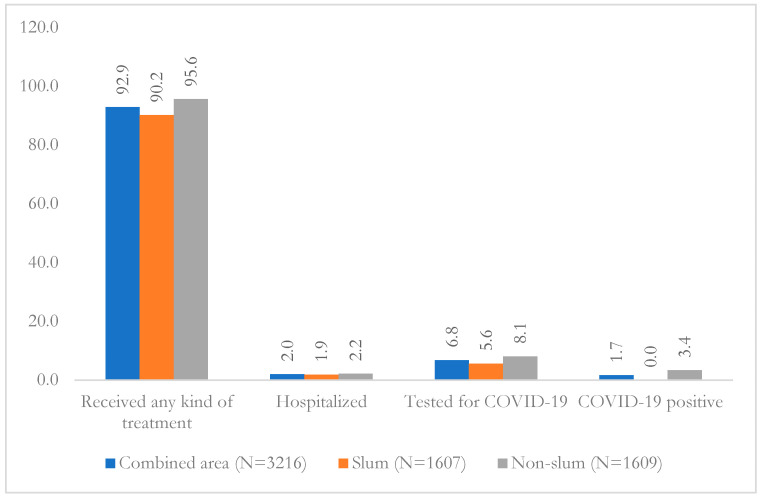
Treatment, hospitalization, and COVID-19 test (RT-PCR) among reported COVID-like symptoms in slum, non-slum, and combined area.

**Table 1 healthcare-11-01444-t001:** Distribution of population ^1^ reported COVID-like symptoms and seroprevalence data matched with COVID-like symptoms by socio-demographic characteristics, slum, non-slum, and combined area.

	Reported COVID-like Symptoms			Seroprevalence Data Matched with COVID-like Symptoms		
Variable	Combined Area	Slum	Non-Slum	Combined Area	Slum	Non-Slum
	(n = 10,050)	(n = 5468)	(n = 4582)	(n = 3205)	(n = 2232)	(n = 973)
	%	%	%	%	%	%
**City**						
Dhaka	53.9	60.7	45.7	81.1	85.1	72.2
Chattogram	46.1	39.3	54.3	18.8	14.9	27.8
**Area**						
Slum	54.4			69.6		
Non-slum	45.6			30.4		
**Age group (years)**						
10–17	18.9	19.9	17.7	24.0	25.8	19.9
18–59	73.6	73.8	73.5	69.4	68.6	71.0
60 or more	7.5	6.3	8.9	6.6	5.6	9.1
**Sex**						
Male	41.3	45.0	36.8	43.2	44.9	39.3
Female	58.7	55.0	63.2	56.8	55.1	60.7
**Education (years of schooling)**						
None	20.8	34.8	4.1	26.1	35.4	4.7
1–4	15.4	22.8	6.5	18.0	23.4	5.6
5–9	33.4	34.5	32.1	33.7	33.6	33.9
10 or more	30.4	7.8	57.3	22.2	7.5	55.8
**Occupation**						
Not-working/unemployed	9.9	13.1	6.1	14.1	17.0	7.4
Service	15.0	16.9	12.8	14.6	14.6	14.5
Business	8.4	7.2	9.9	8.1	7.4	9.8
Self-employed	11.1	18.2	2.6	10.4	13.7	3.0
House-wife	32.4	25.3	41.0	26.3	22.9	34.0
Student	21.6	17.7	26.2	25.1	23.0	29.7
Others	1.6	1.7	1.5	1.4	1.3	1.6
**Income (BDT)**						
<15,000	28.0	48.4	3.7	33.7	46.9	3.4
15,000–19,000	13.3	22.1	2.7	14.6	19.5	3.3
20,000 or more	58.7	29.5	93.6	51.7	33.6	93.3
**Income reduced**						
No	27.7	28.5	26.8	74.8	73.7	77.1
Yes	72.3	71.5	73.2	25.2	26.3	22.9
**Total**	**100.0**	**100.0**	**100.0**	**100.0**	**100.0**	**100.0**

^1^ Age 10 or more years; BDT: Bangladeshi Taka.

**Table 2 healthcare-11-01444-t002:** Reported COVID-like symptoms as well as seroprevalence data matched with COVID-like symptoms (percent) by socio-demographic characteristics, slum, non-slum, and combined area.

Variable	Reported COVID-like Symptoms (Percent)			Seroprevalence Data Matched with COVID-like Symptoms (Percent)		
	Combined Area (n = 10,050)	Slum (n = 5468)	Non-Slum (n = 4582)	Combined Area (3205)	Slum (n = 2232)	Non-Slum (n = 973)
**City**						
Dhaka	30.0	25.8	36.7	26.0	22.7	34.9
Chattogram	34.3	34.9	33.7	25.5	24.0	27.3
**Age group (years)**						
10–17	27.3	24.7	30.8	19.2	16.9	26.3
18–59	33.1	30.6	36.1	28.3	25.3	35.2
60 or more	33.3	30.3	36.0	25.3	22.4	29.5
**Sex**						
Male	32.8	28.4	39.1	26.3	22.7	35.9
Female	31.5	30.2	32.8	25.6	23.1	30.8
**Education (years of schooling)**						
None	26.1	26.2	24.7	20.3	20.0	25.5
1–4	29.9	30.0	29.3	23.9	22.8	35.2
5–9	31.7	31.2	32.4	26.7	25.5	29.3
10 or more	37.4	33.7	38.0	32.5	24.4	35.0
**Occupation**						
Not-working/unemployed	25.8	21.8	36.1	18.0	14.8	34.7
Service	36.7	32.2	43.9	32.5	28.7	41.1
Business	38.8	35.0	42.0	33.8	30.3	40.0
Self-employed	31.0	30.4	36.4	28.7	27.8	36.9
Housewife	32.2	31.2	33.0	27.2	23.2	33.2
Student	28.7	25.5	31.3	21.9	19.5	26.3
Others	36.6	38.3	34.3	20.0	27.6	6.2
**Income (BDT)**						
<15,000	30.8	30.4	36.9	24.2	23.8	36.4
15,000–19,000	30.1	30.0	30.6	24.1	24.3	21.9
20,000+	33.0	27.2	35.2	27.6	20.9	33.0
**Income reduced**						
No	30.2	28.0	33.0	25.9	22.7	32.8
Yes	32.7	29.9	35.9	26.1	23.5	32.7
**Total**	**32.0**	**29.4**	**35.1**	**25.9**	**22.9**	**32.8**

BDT: Bangladeshi Taka.

**Table 3 healthcare-11-01444-t003:** Seroprevalence and COVID-like symptoms of the same individuals in slum, non-slum and combined area.

Seroprevalence	Reported COVID-like Symptoms		Chi^2^ (df) *p*-Value
	Yes n (%)	No n (%)	
**Combined (N = 3205)**			
Seropositive	831 (37.77)	1369 (62.23)	13.61 (1)
Seronegative	312 (31.04)	693 (68.96)	<0.001
**Slum (N = 2232)**			
Seropositive	512 (31.98)	1089 (68.02)	1.49 (1)
Seronegative	185 (29.32)	446 (70.68)	0.222
**Non-slum (N = 973)**			
Seropositive	319 (53.26)	280 (46.74)	34.54 (1)
Seronegative	127 (33.96)	247 (66.04)	<0.001

**Table 4 healthcare-11-01444-t004:** Multivariate logistic regression estimates of reported COVID-like symptoms by selected socio-demographic covariates.

Factors	Reported COVID-like Symptoms					
	Combined ^1^		Slum Area ^2^		Non-Slum Area ^2^	
	aOR (95% CI)	*p*-Value	aOR (95% CI)	*p*-Value	aOR (95% CI)	*p*-Value
**City**						
Dhaka	1.00		1.00		1.00	
Chattogram city	1.13 (1.03, 1.23)	0.007	1.44 (1.27, 1.63)	<0.001	0.86 (0.76, 0.97)	0.016
**Area**						
Slum	1.00		-		-	
Non-slum	1.19 (1.05, 1.36)	0.009	-		-	
**Age group (years)**						
10–17	1.00		1.00		1.00	
18–59	0.99 (0.84, 1.17)	0.948	1.08 (0.86, 1.35)	0.490	0.91 (0.69, 1.18)	0.464
60 or more	1.13 (0.90, 1.41)	0.296	1.25 (0.91, 1.71)	0.174	1.01 (0.70, 1.44)	0.972
**Sex**						
Female	1.00		1.00		1.00	
Male	0.99 (0.88, 1.10)	0.820	0.89 (0.77, 1.03)	0.121	1.16 (0.97, 1.39)	0.098
**Education (years of schooling)**						
None	1.00		1.00		1.00	
1–4	1.27 (1.09, 1.48)	0.002	1.28 (1.08, 1.52)	0.005	1.34 (0.87, 2.07)	0.178
5–9	1.30 (1.14, 1.48)	<0.001	1.25 (1.07, 1.45)	0.004	1.54 (1.07, 2.20)	0.019
10 or more	1.59 (1.36, 1.86)	<0.001	1.41 (1.11, 1.80)	0.005	1.91 (1.34, 2.12)	<0.001
**Occupation**						
Not-working/unemployed	1.00		1.00		1.00	
Service	1.52 (1.26, 1.84)	<0.001	1.45 (1.13, 1.85)	0.003	1.41 (1.02, 1.93)	0.036
Business	1.58 (1.28, 1.96)	<0.001	1.77 (1.32, 2.38)	<0.001	1.26 (0.90, 1.75)	0.177
Self-employed	1.34 (1.09, 1.64)	0.004	1.37 (1.08, 1.76)	0.011	1.16 (0.72, 1.86)	0.532
Housewife	1.19 (0.99, 1.43)	0.069	1.25 (0.97, 1.60)	0.080	1.03 (0.75, 1.41)	0.839
Student	1.00 (0.84, 1.22)	0.966	1.06 (0.82, 1.37)	0.649	0.86 (0.61, 1.21)	0.394
Others	1.46 (1.02, 2.09)	0.040	1.85 (1.16, 2.95)	0.010	0.93 (0.53, 1.65)	0.809
**Income (BDT)**						
<15,000	1.00		1.00		1.00	
15,000–19,000	0.90 (0.78, 1.04)	0.159	0.91 (0.78, 1.06)	0.221	0.71 (0.43, 1.18)	0.186
20,000 or more	0.84 (0.74, 0.95)	0.006	0.84 (0.73, 0.97)	0.016	0.88 (0.63, 1.21)	0.432
**Income reduced**						
Yes	1.00		1.00		1.00	
No	1.14 (1.03, 1.26)	0.008	1.11 (0.97, 1.27)	0.119	1.14 (0.99, 1.32)	0.062
**Summary statistics**	No. of obs: 10050 LR chi^2^ (16): 152.37, *p* < 0.001 −log likelihood: 6223.85 Pseudo R^2^: 0.0121		No. of obs: 5468 LR chi^2^ (16): 106.38, *p* < 0.001 −log likelihood: 3258.23 Pseudo R^2^: 0.0161		No. of obs: 4582 LR chi^2^ (16): 74.74, *p* < 0.001 −log likelihood: 2932.50 Pseudo R^2^: 0.0126	

aOR: adjusted odd ratio; CI: confidence interval; BDT: Bangladeshi Taka. ^1^ The model includes city, are, age groups, sex, education, occupation, income, and income reduced. ^2^ The model includes city, age groups, sex, education, occupation, income, and income reduced.

**Table 5 healthcare-11-01444-t005:** Multivariate logistic regression estimates of seropositive data matched with COVID-like symptoms by selected socio-demographic covariates.

Factors	Seroprevalence Data Matched with COVID-like Symptoms					
	Combined ^1^		Slum Area ^2^		Non-Slum Area ^2^	
	aOR (95% CI)	*p*-Value	aOR (95% CI)	*p*-Value	aOR (95% CI)	*p*-Value
**City**						
Dhaka	1.00		1.00		1.00	
Chattogram	0.88 (0.72, 1.09)	0.259	1.05 (0.79, 1.40)	0.724	0.75 (0.55, 1.03)	0.076
**Area**						
Slum	1.00		-		-	
Non-slum area	1.67 (1.31, 2.14)	<0.001	-		-	
**Age group (years)**						
10–17	1.00		1.00		1.00	
18–59	1.21 (0.90, 1.64)	0.204	1.33 (0.92, 1.92)	0.130	0.96 (0.54, 1.69)	0.882
60 or more	1.22 (0.80, 1.87)	0.363	1.37 (0.79, 2.37)	0.261	0.87 (0.39, 1.94)	0.740
**Sex**						
Female	1.00		1.00		1.00	
Male	0.98 (0.80, 1.20)	0.849	0.86 (0.68, 1.09)	0.215	1.31 (0.90, 1.91)	0.158
**Education (years of schooling)**						
None	1.00		1.00		1.00	
1–4	1.28 (0.98, 1.68)	0.074	1.20 (0.89, 1.60)	0.233	1.75 (0.71, 4.30)	0.223
5–9	1.26 (0.99, 1.60)	0.063	1.23 (0.94, 1.60)	0.129	1.24 (0.59, 2.60)	0.569
10 or more	1.32 (0.98, 1.78)	0.071	1.13 (0.73, 1.72)	0.584	1.52 (0.74, 3.11)	0.254
**Occupation**						
Not-working/unemployed	1.00		1.00		1.00	
Service	1.80 (1.28, 2.52)	0.001	1.93 (1.28, 2.91)	0.002	1.29 (0.67, 2.47)	0.450
Business	1.90 (1.30, 2.78)	0.001	2.29 (1.42, 3.68)	0.001	1.15 (0.58, 2.29)	0.681
Self-employed	1.76 (1.22, 2.52)	0.002	1.90 (1.26, 2.87)	0.002	1.25 (0.48, 3.22)	0.647
Housewife	1.32 (0.94, 1.85)	0.105	1.32 (0.88, 1.99)	0.182	1.16 (0.61, 2.22)	0.654
Student	1.12 (0.81, 1.57)	0.490	1.34 (0.90, 2.01)	0.154	0.70 (0.35, 1.42)	0.327
Others	0.94 (0.43, 2.06)	0.873	1.85 (0.77, 4.46)	0.170	0.14 (0.02, 1.15)	0.068
**Income (BDT)**						
<15,000	1.00		1.00		1.00	
15,000–19,000	0.94 (0.72, 1.22)	0.630	0.99 (0.75, 1.30)	0.927	0.51 (0.17, 1.56)	0.239
20,000 or more	0.85 (0.69, 1.07)	0.163	0.83 (0.65, 1.06)	0.137	0.91 (0.43, 1.91)	0.808
**Income reduced**						
Yes	1.00		1.00		1.00	
No	0.97 (0.80, 1.17)	0.746	0.94 (0.74, 1.20)	0.628	1.01 (0.72, 1.40)	0.966
**Summary statistics**	No. of obs: 3205 LR chi^2^ (16): 87.48, *p* < 0.001 −log likelihood: 1790.50 Pseudo R^2^: 0.0238		No. of obs: 2232 LR chi^2^ (16): 44.45, *p* < 0.001 −log likelihood: 1179.79 Pseudo R^2^: 0.0185		No. of obs: 973 LR chi^2^ (16):30.17, *p* < 0.05 −log likelihood: 600.58 Pseudo R^2^: 0.0245	

aOR: adjusted odd ratio; CI: confidence interval; BDT: Bangladeshi Taka. ^1^ The model includes city, are, age groups, sex, education, occupation, income, and income reduced. ^2^ The model includes city, age groups, sex, education, occupation, income, and income reduced.

## Data Availability

To protect the identification of the participants, some restrictions do apply to the primary data. These data can be made available from the Ethics Committees (ERC/RRC) at the International Centre for Diarrhoeal Disease Research, Bangladesh (icddr,b) for researchers who meet the criteria for access to confidential data; please contact the Head of Research Administration at the icddr,b (Armana Ahmed: aahmed@icddrb.org). For more details, please visit to http://www.icddrb.org/policies.

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
