# Peer review of "Factors Associated with Reported COVID-like Symptoms and Seroprevalence Data Matched with COVID-like Symptoms in Slums and Non-Slums of Two Major Cities in Bangladesh"

_healthcare, 2023, doi:10.3390/healthcare11101444_

Round 1

Reviewer 1 Report

"Factors associated with reported COVID-like symptoms and seroprevalence data matched with COVID-like symptoms: Evidence from selected slums and non-slums of Dhaka and Chattogram cities in Bangladesh" manuscript by Razzaque et al., describe SARS CoV2-seropositivity among slums and non slums area during COVID19 pandemic, prior to the adoption of vaccine. 

While the information was interesting, below are some input:

Methods:

1. Please state ethical permission as needed

2. Subject selection: while it was written down that subjects were selected, please add how did the selection done, and potential bias that might occur.

3. Data analysis and results

a. For IgM and IgG to N protein of SARS CoV2, it was not mentioned to which antibody did the seropositivity was measured from, as IgM and IgG reads different condition. While discussing symptoms, it was commonly agreed that COVID19 symptoms commonly were > 1 symptoms, and in flu-related symptoms, some were hard to distinguished, such as fever, sore throat, or joint-ache, etc. Please describe how to avoid the bias.

b. In general, there were so many tables that invite confusion. Authors are suggested to simplify the tables, without minimizing the information. Numbers described were often hard to be connected to the tables, as it was not written which table was being described.

c. Despite the differences in percentage, it is suggested to run descriptive analysis to observe the statistical differences

d What would be result of Z score be adopted to the analysis was not described

e. It is advisable that the authors run crude and adjusted logistic regression by including  variables with potential bias as indicated by descriptive table. It was unclear in the method to which variables was/ were the adjustment done.

Reviewer 2 Report

This article reported factors associated with reported COVID-like symptoms and seroprevalence data matched with COVID-like symptoms, which is an interesting and important topic today. However, there are some minor concerns in this article that need to be addressed.

1.     Please include the WHO link in line 42.

2.     Please include the exclusion criteria in the method section.

3.     Please include an appropriate reference at the end of sentence.

“Typical symptoms of common cold are usually observed in mild cases, but severe case often requires hospitalization, long hospital stay, and may result in multiorgan failure and increased mortality.”

“In countries with large populations, although having more numbers of both cases and deaths, older people with various health complications suffered the most.”

4.     Was there a significant difference between seropositive individuals and the areas? Please include the P value in text and related table.

5.     In the discussion section, after comparing other studies with this study, no conclusions and possible reasons were explained. Please rewrite the discussion.

6.     Please change “et al.” to “et al.” in the discussion section.

7.     For better understanding, please explain the meaning of the number inside the parentheses in total row of the table 1 and 2.

8.     Please include the strengths of this study.

Minor editing of English language required.

Reviewer 3 Report

"Factors associated with reported COVID-like symptoms and seroprevalence data matched with COVID-like symptoms: Evidence from selected slums and non-slums of Dhaka and Chattogram cities in Bangladesh" manuscript by Razzaque et al., focus on SARS CoV2-seropositivity among slums and non-slums area during COVID19 pandemic. The information is useful but I have following suggestions.

1. In a large-sample cohort study like the present one methods section should be explained elaborately.

2. Too many tables. Suggestion is to simplify and combine them or if possible, authors can provide figures with statistical significance.

3. It will be helpful for the readers if P values are mentioned in the text.

4. The authors can try to make the title more clear.

Thank you.

Minor Proofreading is required 

Round 2

Reviewer 1 Report

The authors have provided revision as requested and add clearance to the manuscript in describing what was observed during early pandemic in Bangladesh.

Author Response

Response to the reviewer 1

The authors have provided revision as requested and add clearance to the manuscript in describing what was observed during early pandemic in Bangladesh.”

Authors response: Thank you very much for your valuable time and feedbacks.
